# Emission factors and light absorption properties of brown carbon from household coal combustion in China

Jianzhong Sun[1, 2, 5], Guorui Zhi[2*], Regina. Hitzenberger[4], Yingjun Chen[1, 3*], Chongguo Tian[1], Yayun Zhang[2, 6], Yanli Feng[7], Miaomiao Cheng[2], Yuzhe Zhang[2, 6], Jing Cai[8], Feng Chen[7], Yiqin Qiu[7], Zhiming Jiang[7], Jun Li[8], Gan Zhang[8], Yangzhi Mo[8]

[1]Key Laboratory of Coastal Environmental Processes and Ecological Remediation, Yantai Institute of Coastal Zone Research, Chinese Academy of Sciences, Yantai 264003, China
[2]State Key Laboratory of Environmental Criteria and Risk Assessment, Chinese Research Academy of Environmental Sciences, Beijing 100012, China
[3]State Key Laboratory of Pollution Control and Resources Reuse, Key Laboratory of Cities' Mitigation and Adaptation to Climate Change, College of Environmental Science and Engineering, Tongji University, Shanghai 200092, China
[4]University of Vienna, Faculty of Physics, Boltzmanngasse 5, 1090 Vienna, Austria
[5]University of Chinese Academy of Sciences, Beijing, 100049, China
[6]College of Chemical Engineering, China University of Petroleum, Beijing 102249, China
[7]Institute of Environmental Pollution and Health, School of Environmental and Chemical Engineering, Shanghai University, Shanghai 200444, China
[8]State Key Laboratory of Organic Geochemistry, Guangzhou Institute of Geochemistry, Chinese Academy of Sciences, Guangzhou 510640, China

*Correspondence to*: Guorui Zhi (zhigr@craes.org.cn) and Yingjun Chen (yjchentj@tongji.edu.cn)

**Abstract.** Brown carbon (BrC) draws increasing attention due to its effects on climate and other fields. In China, household coal burned for heating/cooking purposes releases huge amounts of carbonaceous particles every year; however, BrC emissions have rarely been estimated in a persuasive manner due to the unavailable emission characteristics. Here 7 coals jointly covering geological maturity from low to high were burned in 4 typical stoves at both chunk and briquette styles. The optical integrating sphere (IS) method was applied to measure the emission factors (EFs) of BrC and black carbon (BC) via an iterative process using the different spectral dependence of light absorption for BrC and BC and using humic acid sodium salt (HASS) and carbon black (CarB) as reference materials. It is found that (i) the average EFs of BrC for anthracite coal chunks and briquettes are (1.08±0.80) g/kg and (1.52±0.16) g/kg, respectively, and those for bituminous coal chunks and briquettes are (8.59±2.70) g/kg and (4.01±2.19) g/kg, respectively, reflecting a more significant decline of BrC EFs for bituminous coals than for anthracites due to briquetting, (ii) the BrC EF peaks at the middle of coal's geological maturity, displaying a bell shaped curve between EF and volatile matter ($V_{daf}$), (iii) the calculated BrC emissions from China's residential coal burning amounted to 592 Gg (1 Gg=$10^9$ g) in 2013, which is nearly half of China's total BC emissions, (iv) absorption Ångström exponent (AAEs) of all coal briquettes are higher than those of coal chunks, indicating that the measure of coal briquetting increases the BrC/BC emission ratio and thus offsets some of the climate cooling effect of briquetting, and (v) in the scenario of current household coal burning in China, solar light absorption by BrC (350-850 nm in this study) accounts for more than a quarter (0.265) of the total absorption. This implies the significance of BrC to climate modeling.

## 1 Introduction

The past decade saw increased interest in brown carbon (BrC) due to its effects on atmospheric chemistry, air quality, health, and particularly climate (Andreae and Gelencsér, 2006; Saleh et al., 2014; Forrister et al., 2015; Laskin et al., 2015). BrC refers to the fraction of organic carbon (OC) that can absorb light (Yan et al., 2014; Zhi et al., 2015; Jo et al., 2016; Wang et al., 2016). Compared with black carbon (BC) that has usually been considered the strongest light-absorbing aerosol carbon with an absorption Ångström exponent (AAE) of around 1.0, light absorption by BrC is weaker, but more strongly wavelength dependent (Kirchstetter and Novakov, 2004; Hoffer et al., 2006; Cai et al., 2014). In other words, the light absorption efficiency of BrC increases more than that of BC toward short wavelengths. Recent advances in BrC research revealed its abundances and properties in a number of regions and highlighted the importance of including BrC in the accurate modeling of aerosol radiative forcing (RF) (Mohr et al., 2013; Zhang et al., 2013a; Chakrabarty et al., 2014; Du et al., 2014; Kirillova et al., 2014; Forrister et al., 2015; Liu et al., 2015; Washenfelder et al., 2015; Cheng et al., 2016). For example, Feng et al. (2013) used a global chemical transport model and a radiative transfer model, finding that the strongly absorbing BrC contributes up to $+0.25$ $W \cdot m^{-2}$ or 19% of the absorption by anthropogenic aerosols; meanwhile the RF at the top of the troposphere may change from $-0.08$ $W \cdot m^{-2}$ (cooling) to $0.025$ $W \cdot m^{-2}$ (warming) in some areas when BrC data are considered in the RF model. Park et al. (2010) combined a 3-D global chemical transport model (GEOS-Chem) with aircraft/ground based observations, finding that the averaged RFs of BrC aerosol were $0.43$ $W \cdot m^{-2}$ at the surface and $0.05 W \cdot m^{-2}$ at the top of atmosphere (TOA), respectively, both of which accounted for more than 15% of their respective total RFs ($2.2$ $W \cdot m^{-2}$ and $0.33$ $W \cdot m^{-2}$) for absorbing aerosols.

The sources of BrC can be defined into two categories: primary emission and secondary formation. The former relates to the incomplete (even smoldering) combustion of either fossil fuels (coal, petroleum and natural gas, etc.) or biomass/biofuels (wood, agricultural residues and bio-ethanol, etc.) during which BrC is generated and released into the atmosphere as pollutants (Liu et al., 2014; Oris et al., 2014; Washenfelder et al., 2015; Zhi et al., 2015); the latter involves complex chemical reactions taking place in the atmosphere between various precursors, forming secondary organic aerosols (SOAs), some of which are light-absorbing (Laskin et al., 2014; Lee et al., 2014; Smith et al., 2014; Tóth et al., 2014; Martinsson et al., 2015; Yan et al., 2015; Zhao et al., 2015). The SOA precursors of anthropogenic origin are usually dominated by hydrocarbons like aromatics and aliphatics, and those of natural origin are mainly biogenic volatile organic compounds (BVOCs) like isoprene and monoterpene (Updyke et al., 2012; Faiola et al., 2014; Fu et al., 2014; Liu et al., 2014). Examples on characterization of the two BrC source categories are relatively rare, which is unfavorable to the accurate understanding of BrC in terms of sources and effects (Laskin et al., 2015; Zhi et al., 2015). In addition, there is a more pressing need to characterize BrC emissions from sources related to human activities, so as to seek direct insight into the influence of anthropogenic emissions on global change.

In China, coal plays a dominant role in the energy structure. In 2013, coal consumption reached 4300 Tg (1 Tg=$10^{12}$ g), accounting for approximately 70% of China's total primary energy, 93 Tg of which was burned for household

heating/cooking purposes (NBSC, 2014). Because burning raw coal in residential cooking/heating stoves has the potential to release pollutants that consist of up to 10% of the fuel mass due to poor combustion conditions and control facilities (Zhang and Smith, 2007), huge emissions of carbonaceous particles including OC, BrC, and BC are expected in this sector. Our previous studies have shown that the emission factors (EFs) of BC for residential coal burning are closely related to coal's

ranks (bituminous or anthracite) and processed styles (raw coal chunk or coal briquette), but never addressed BrC that is released concurrently with BC (Chen et al., 2006, 2009; Zhi et al., 2008, 2009). Meanwhile, the optical properties of BrC from coal combustion have almost never been addressed by researchers around the world, possibly because studies regarding BrC emission have focused preferentially on the observed physical or chemical properties, particularly optical absorption (e.g., AAE) of ambient aerosols for an overall characterization of radiative impacts of BrC in the atmosphere in a certain

region or from specific burning activities (Chakrabarty et al., 2013; Feng et al., 2013; Lack and Langridge, 2013; Wu et al., 2013; Zhang et al., 2013a; Zheng et al., 2013; Du et al., 2014; Washenfelder et al., 2015; Yan et al., 2015; Zhao et al., 2015; Cheng et al., 2016). These are not conducive to the understanding of China's primary BrC emission characteristics, especially BrC from residential sector, the largest contributor of primary carbonaceous particles in China (Streets et al., 2013; Cai et al., 2014; Zhi et al., 2015).

The general motivation of this study is to investigate the emissions and optical characteristics of BrC emitted by China's household coal burning. A group of coals jointly covering geological maturity from low to high were burned in various stoves at both chunk and briquette styles, accompanied by collecting particulate emissions on quartz fiber filters (QFFs). The optical integrating sphere (IS) approach was used to distinguish BrC from BC on the filters (Hitzenberger et al., 1996; Wonaschütz et al., 2009; Montilla et al., 2011), followed by the calculation of EFs of BrC and other particulate components.

The calculated BrC emissions and light-absorbing contributions add to the importance of China's household coal burning in both climate and air quality.

## 2 Experimental Section

### 2.1 Coals and Stoves

Seven coals were prepared in the present study (Table 1). These coals cover a wide range of geological maturity and can

be classified into one anthracitic coal (AN), one semi-anthracite coal (SA), one low volatile bituminous coal (LVB), one medium-volatile bituminous coal (MVB), and three high-volatile bituminous coals (HVB). Each coal was prepared into two styles: raw-coal chunk and honey-comb briquette. The raw-coal chunks were 3-6 cm in size and the honeycomb briquettes were made by intermixing coal powder with clay (25%) into a 12-hole column, 6 cm in height and 9.5 cm in diameter (Chen et al., 2005, 2015a; Zhi et al., 2008).

Four household coal-stoves were selected to represent the most popular stove patterns used in northern China: one of them is specifically for honeycomb briquettes (WJ) and the other three are for raw-coal chunks (SC, HD, and LW). Detailed information on these stoves regarding shape, size, and characteristic structure is presented in Supporting Information (Figure

S1) and will be described here briefly. The briquette stove WJ and chunk stove SC are of traditional style widely used especially in past decades in China's households for heating rooms through direct thermal radiation. HD and LW are actually mini-boilers of low pressure type used for heating rooms by heated water circulating through a piping system. Compared to HD, the LW stove has an additional iron baffle vertically fixed before the flue pipe so as to lengthen the time of heat exchange between hot flue gas and circulating water.

## 2.2 Coal Combustion and Sample Collection

Since the briquettes of 7 coals were only burned in stove WJ and the chunks of 7 coals all were burned in the 3 chunk stoves (SC, HD, LW) one by one, there were a total of 28 coal/stove combinations for the emission test. At first, two or three anthracite-briquettes (ca. 600 g each) were ignited outdoors by solid alcohol until the carbon burning stage of coal was reached to minimize the interferences of igniting alcohol and anthracite briquettes in subsequent coal tests. Then the stove was moved into the preset position of the burning-sampling system. A batch of coal briquettes (1-3) or chunks (0.5-3 kg) were put into the stove and were ignited from the bottom by pre-burned anthracite briquettes. When the combustion began to fade (the first burning cycle, 1-2 h), a new batch of test coal briquettes or chunks were added into the stoves until being burned completely (the second burning cycle, 1-2 h). Some coals (especially AN and SA) were burned for a third cycle (1-2 h) to ensure enough particle sampling.

Samples were collected through a diversion-dilution-sampling system (Supporting Information Figure S2). Coal burning emissions were released to the air through a 3m-long iron-chimney. A small flue gas stream (ca. 1-3 l/min) was diverted from the chimney mainstream into a FPS-4000 (Dekati) dilution device. The side opening of the chimney for stream diversion was 50 cm above the stoves. The dilution ratio in this study ranged from 30 to 180, depending on the envisaged emission intensity of each combination as well as on burning conditions. For example, emissions from the two anthracites (AN, SA) were less diluted by clean air than those from bituminous coals due to a lower emission concentration expected for the former than for the latter. There were 6 outlets at the end of the FPS-4000, which could be used to connect to different sampling/monitoring instruments including at least a particle sampler (Φ90mm Pallflex QFFs) for future BrC determination by the IS approach (Wonaschütz et al., 2009). In addition, the flue gas temperature, flow velocity, and composition were all simultaneously monitored by a digital thermocouple, a KURZ flowmeter, and a flue gas analyzer, respectively, throughout sampling so that the combustion processes could be characterized as fully as possible.

For each coal/stove combination, sampling started when the first batch of coal was put into the stove and ended when combustion was over (Zhi et al., 2008; Chen et al., 2015a, b). QFFs used for sample collection were baked at 450 ºC in a muffle furnace for 6 h to get rid of any organics adsorbed on the filters. The combustion experiment for each coal/stove combination was done twice to check for reproducibility. Blanks were also tested to correct for the influences of whole procedures and anthracite briquettes used for initial igniting.

It should be noted that we chose to perform our study through lab-experiments rather than real field tests is because the former is easier to be controlled and repeatable than the latter, which allowed us to test the briquetting effects or coal rank

effects by fixing other conditions (the same 7 coals, identical combustion manipulation, and consistent sampling system) (Jenkins et al., 1996; Roden et al., 2009; Zhang et al., 2011; Jetter et al., 2012). However, more and more studies suggested that lab studies may fail to simulate high emissions and be difficult to capture high variations in real field (Roden et al., 2006; Johansson et al., 2008; Christian et al., 2010). In this sense, future study is proposed to go to real field manner.

## 2.3 Measurement of BrC with IS Method

IS method was utilized in this study to separate the contributions of BrC and BC in terms of light absorption. A 150 mm integrating sphere (manufactured by Labsphere, Inc) was built in a UV-Vis-NIR spectrophotometer (Perkin Elmer Lambda 950). The sphere is coated internally with Polytetrafluoroethylene (PTFE), which reflects > 99% of the incident light in the wavelength range of 0.2-2.5 µm (Wonaschütz et al., 2009). Using the full-scan mode, we scanned through the wavelength range of 350-850 nm to measure the light absorption of samples. A transparent quartz cuvette was specially customized and placed in the center of the sphere to hold filter samples for optical measurement. Inside the cuvette was 3 ml of 1:1 mixture of acetone and a 80:20 mixture of water/isopropanol in which a filter punch (rectangle punch, 30×8 mm) could be immersed. Around the quartz cuvette was a specially customized cuvette holder coated with PTFE to hold the quartz cuvette in the sphere center. The sketch diagram of the IS measurement principle is shown in Figure 1.

As discussed by Hitzenberger and Tohno (2001) and Wonaschütz et al. (2009), samples are put into the liquid mixture for the following consideration. Non-absorbing coatings on light absorbing particles lead to appreciably enhanced absorption efficiencies. In the liquid, soluble coatings are removed. Typical insoluble coatings of aerosol particles (mainly organic material) have refractive indices around 1.4 (D'Almeida et al., 1989), which is similar to that of the liquid mixture (1.35). The resulting relative refractive index is small enough (1.04) to render the absorption enhancement by the coating negligible.

Reference materials need to be used as calibration standards to link the measured optical signals to the amounts of absorbing materials. Available reference materials were usually carbon black (CarB) (e.g., Elftex 570, Cabot Corporation) for BC and humic acid sodium salt (HASS) (e.g., Acros Organics, no. 68131-04-4) for BrC (Heintzenberg, 1982; Reisinger et al., 2008; Wonaschütz et al., 2009). For example, in the study of Medalia et al. (1983), CarB was used as the proxy of BC in diesel exhaust and in the study of Wonaschütz et al. (2009), HASS was used as proxies for BrC from wood combustion. We carry over this philosophy to the current study, with an assumption that BC and BrC in household coal smoke have the same light-absorbing properties of CarB and HASS, respectively. Consequently it is not surprising that the results obtained here are probably different from others in literature reporting BC and/or BrC using other measurement techniques (e.g., thermal/optical method or aethalometer) (Chen et al., 2006; Zhi et al., 2008, 2009; Shen et al., 2013, 2014; Aurell and Gullett, 2013) or reference materials (e.g., Fulvic acid, humic acid or HULIS) (Duarte et al. 2007; Lukács, et al. 2007; Baduel et al. 2009, 2010). Even if this assumption is anyway not perfect because the properties of CarB and HASS may never be completely the same as BC and BrC released from either wood, diesel, or coal, researchers can still use them to link and compare the emission characteristics of BC and BrC from various sources.

Calibration curves were obtained for a series of CarB masses of 1.5-90 µg and HASS masses of 3-240 µg at wavelengths of 650 nm and 365 nm. The reason why we used two wavelengths is the different spectral dependence of the absorptive characteristics of BrC and BC, based on which a gradual separation of BrC from BC could be realized through iteration procedures. Different from CarB that is composed of almost pure carbon, HASS contains only 47% carbon by weight. For this reason, all measured HASS equivalent values based on such a calibration curve must be multiplied by 0.47 to obtain real BrC. The separation method of BC and BrC was generally similar to that by Wonaschütz et al. (2009), except that 405 nm was replaced by 365 nm because 365 nm is more preferred by researchers in BrC research and because the strong spectral dependence of absorption by BrC enables a better separation of the contributions of BC and BrC at this wavelength (Zhang et al., 2013a; Du et al., 2014; Yan et al., 2014, 2015; Zhi et al., 2015). Figure 2 shows the calibration curves for BrC and BC at both 365 nm and 650 nm. At 650 nm, HASS gives only about 3% of the signal of an equal mass of CarB, yet at 365 nm, HASS gives 24% of the signal of an equal mass of CarB. With the 4 calibration curves in Figure 2, filter samples were analyzed for BrC and BC with the IS method.

## 2.4 Calculation Methods

Details of the methods for calculating EFs (for BrC and BC), absorption Ångström exponent (AAEs), wavelength dependent BrC contribution to light absorption ($f_{BrC}(\lambda)$) and average BrC contribution to solar light absorption ($F_{BrC}$) in the range of 350-850 nm are given in the Supporting Information.

## 3 Results and Discussion

### 3.1 Influence of Coal Briquetting on the EFs of BrC

The calculated emission factors of BrC and BC for the coal/stove combinations are presented in Table 2. Based on Table 2, Figure 3 is derived to show the influence of coal briquetting on the EFs for BC and BrC. Turning coal powder into briquette is considered one of the effective approaches to reduce emissions of many pollutants, and has been vigorously promoted by the Chinese government since its 9th five-year-plan period (1995-2000) (Cheng et al., 1998; Chen et al., 2009; Zhi et al., 2009). Our previous studies showed that briquetting reduces emissions of BC and some other pollutants (e.g., OC, particle matter (PM)) drastically, making briquetting possibly an option for both climate and environmental protection (Zhi et al., 2009; Shen et al., 2014). The effect of reducing BC emissions is seen also in the present study. As shown in Figure 3a and Table 2, the average EFs of BC for chunk- and briquette- anthracites are (0.43±0.23) g/kg and (0.21±0.16) g/kg, respectively, indicating a more than 50% drop due to briquetting of anthracites; meanwhile the average EFs of BC for chunk- and briquette- bituminous coals are (7.85±2.00) g/kg and (0.56±0.22) g/kg, respectively, reflecting a more than 90% drop due to briquetting of bituminous coals. It is believed that the structure (multihole for ventilation and burning) and composition (including 1/3 clay) of coal briquettes help the complete combustion of coal and thereby less BC is released by the burning of briquettes than that of coal chunks (Bond et al., 2004; Zhi et al., 2009).

Regarding BrC emissions (Figure 3b), no significant decline in EFs is seen for anthracite briquetting (EF is (1.08±0.80) g/kg for chunks, and (1.52±0.16) g/kg for briquettes), but a notable decline in EFs is observed for bituminous coals (EF is (8.59±2.70) g/kg for chunks, and (4.01±2.19) g/kg for briquettes), displaying a reduction of 53% due to briquetting. On the one hand, we are convinced that coal briquetting can generally lower BrC emissions, particularly for the bituminous coals (a

53% reduction) that are more widely used in China than anthracite coals; on the other hand, the magnitude of BrC decrease for bituminous coals is significantly less than that for BC (more than 90% reduction for bituminous coals). The lesser decline of BrC compared to that of BC due to briquetting of bituminous coals may be due to the different formation mechanisms of BrC and BC. Although such mechanisms have never been specifically addressed, evidence regarding the influence of briquetting in polycyclic aromatic hydrocarbons (PAHs) may indirectly contribute to accounting for the difference.

According to Pöschl (2003), BrC aerosols are optically colored organics and thermochemically refractory organics, some of which are polycyclic aromatics. Chen et al. (2015b) observed that the EFs of 16 parent PAHs, 26 nitrated PAHs, 6 oxygenated PAHs, and 8 alkylated PAHs for coal briquettes were higher than those for coal chunks, corroborating the difference in formation mechanisms between BrC and BC. The authors tried a tentative insight into the enhancement of PAH emissions and speculated that PAHs are not affected as much by combustion efficiency as BC and might be more greatly

affected by pyrolytic process instead. According to the authors, PAHs are formed through two inter-linked processes: pyrolysis and pyrosynthesis (Bjorseth and Ramdahl, 1985; Barbella et al., 1990; Bonfanti and Theodosis, 1994; Mastral et al., 1996, 1999, 2000); turning the mixture of coal powder and clay into briquette favors the pyrosynthesis (Chen et al., 2015b). Here with the example of PAHs, we attempt to show that BrC (including PAHs) behaves differently from BC, and that further investigations on the different effects of briquetting on BrC and BC emissions are needed.

In addition to the BC data from IS method here, we also have EC data from thermal/optical reflectance (TOR) carbon analysis method. BC-EC paired data are given in the Supporting Information (Table S1 and Figure S3). Although IS-BC is somewhat higher than TOR-EC in most cases, they are correlated significantly.

## 3.2  The Dominant Role of Coal Ranks in the EFs for BrC

Based on Table 2, the emission factors for BrC and BC from bituminous and anthracite coals burned in the same four

stoves are plotted in Figure 4 for comparing the overall influence of coal's rank on BrC emissions. Each EF for bituminous coal is the average over 5 bituminous coals and similarly each EF for anthracite coal is the average over 2 anthracites. It is very clear that both BrC and BC have higher EFs for bituminous coals than for anthracites, indicating that anthracites are always cleaner than bituminous coals, either for BC or BrC emissions from either briquettes or chunks. This confirms our previous recognition that coal's geological maturity (represented by $V_{daf}$ value) plays a decisive role in the pollutant emission

factors for residential coal burning because emissions from residential stoves are essentially the result of incomplete combustion of volatile matter in coal (Zhi et al., 2008, 2009). The lower combustion efficiency in household stoves leads to markedly incomplete combustion of volatile matter contained in raw coal, which acts as reactant in producing the final

emissions. This also suggests that burning anthracite coals instead of bituminous coals in residential sector results in lower emissions of light-absorbing carbon BrC and BC, which favors climate protection.

It is interesting that although anthracite coals have been found to have lower EFs for BrC than bituminous coals in general, the EFs do not increase monotonically with $V_{daf}$ but rather display a "bell shape". Previous studies proposed a "bell shaped curve" with a maximum EF at $V_{daf}$ around 30% to describe the variation of BC EFs versus coal $V_{daf}$ when coal is burned in household stoves (Zhi et al., 2008, 2009). In this study, the 7 coals, from left to right in Figure 3a and Figure 3b, are arranged for increasing $V_{daf}$. The bell shape profile of BC is maintained, with the coal PDS ($V_{daf}$ =26.25%) having the highest EFs (0.75-1.00 g/kg for briquettes and 10.15 g/kg for chunks; Figure 3a). Meanwhile as shown in Figure 3b, the bell-shape profile reappears for BrC EFs, with EFs for coal briquettes and chunks peaking respectively in coal PDS ($V_{daf}$=26.25%, $EF_{BrC}$=4.71-8.13 g/kg) and SYS ($V_{daf}$ =33.20%, $EF_{BrC}$=11.49 g/kg). The findings of this study indicate that in order to reduce emissions of light absorbing carbon (BC and BrC), the use of middle maturity coal in residential stoves should be minimized. Similar trend was also found by Shen et al. (2013) for the emissions of particle-bound PAHs (the medium volatile bituminous coals are most productive), which prompts us to propose a ban on this type of coals in household purpose.

We take additional advantage of Figure 4, finding that, among the 3 chunk stoves, both $EF_{BrC}$ and $EF_{BC}$ of coal chunks are in the order of LW>SC>HD, regardless of bituminous coal or anthracite coal, reflecting that HD stove performs the best and the LW stove, the worst. As stove SC is of traditional style widely used especially in past decades in China's households for heating rooms through direct radiation of coal-burning whereas the stoves HD and LW are actually household mini-boilers of low pressure type used for heating rooms by a water piping/radiating system (see subsection 2.1), The above order of LW>SC>HD in terms of EFs implies that current transfer from stove type of direct coal combustion radiation (e.g., SC stove) to that of heated water piping/radiating system (e.g., LW and HD stoves) in households does not necessarily leads to a decline of EFs.

A collection of the directly measured BC (EC) emission factors in our previous articles (since 2005) (Chen et al., 2005, 2006, 2015a; Zhi et al., 2008, 2009) and in this study are given in Supporting Information (Table S2). Comparison between previous and current studies shows that the means of the emission factors for either anthracite coals or bituminous coals in either briquette or chunk styles in this study are somewhat higher than those in previous ones; however the key findings in previous studies still stand in this study. For example, bituminous coals or raw chunks usually release more pollutants (including BC) than anthracites or briquettes, and BC emission factor usually peaks in medium-volatile bituminous coal, etc. The differences in reported EFs are generally from a variety of factors, such as stoves, briquetting procedures, combustion manipulations, and even the quantification methods (Chow et al., 2001; Zhi et al., 2009, 2011).

## 3.3 Absorption Ångström Exponent (AAE)

The calculated AAE values for China's residential coal combustion are shown in Figure 5. It is very obvious that AAEs of all coal briquettes are higher than those of coal chunks. For coal-briquettes, AAE values are in the range of 2.11-3.18, with the average of 2.55±0.44, while for coal-chunks, AAE values decline to 0.96-1.73, with an average of 1.30±0.32, which is

nearly a half of that for coal-briquettes. This may be attributed to the higher ratio of $EF_{BrC}/EF_{BC}$ ($R_{BrC/BC}$) for coal briquettes (7.68±3.16, derived from Table 2) than for coal chunks (1.46±0.69, derived from Table 2) in view of the generally higher AAEs for BrC than for BC (Andreae amd Gelencser, 2006; Chen and Bond, 2010; Kirchstetterand Thatcher, 2012; Cai et al., 2014; Yan et al., 2014; Martinsson et al., 2015; Chakrabarty et al., 2016; Wang et al., 2016). This reminds us that although briquetting can reduce both BC and BrC emissions (as shown in Figure 3), BC is far more reduced than BrC, leading to an increased $R_{BrC/BC}$ after briquetting and consequently offsetting the climate cooling effect of briquetting (Zhi et al., 2009). In addition, in Cai et al. (2014)'s study, the AAEs of 10 samples of wheat straw open burning were measured, with an average of 3.02±0.18, much higher than those for coals (chunk or briquette) in this study. The higher OC/TC ratio for biomass burning than for fossil fuel combustion possibly accounts for such a result (Novakov et al., 2005; Cai et al., 2014).

As for the relationship between coal's maturity (represented by $V_{daf}$) and AAE, Figure 5 demonstrates that AAE values do not decrease monotonically with $V_{daf}$ % but display a "U shape" pattern; the minimal AAEs occur in the coals of medium maturity (SYS) (2.11 for briquette and 0.96 for chunk, in Figure 5). It's interesting that this relationship profile is by and large in contrast to the "bell-shape" for the relationship between $EF_{BC}$ and $V_{daf}$ (The maximal $EF_{BC}$ occurred in medium maturity coals) (Zhi et al., 2008, 2009). The mechanism behind this contrast needs further investigation.

## 3.4 Light Absorption by BrC from Household Coal Stoves

Based on the measured EFs in this study and China's yearly consumption of residential coal in the China Energy Statistical Yearbooks (CESYs) (NBSC, 2014), the emissions of BrC and BC from China's coal burning in household stoves can be calculated. According to CESY 2014, 92.90 Tg of coal was used in residential sector in 2013. Assuming that 20% of coal was anthracite, and that 20% of bituminous and anthracite coals were both made into briquettes (Chen et al., 2006; Zhi et al., 2008), the calculated BrC emissions from China's residential sector amounted to 592 Gg (1 Gg=$10^9$ g), which is nearly half of China's BC emissions (Cao et al., 2006a, 2006b, 2011; Wang et al., 2012; Zhang et al., 2013b; Zhang et al., 2015). Chakrabarty et al. (2014) reported a BrC emission of 92 Gg from funeral pyres in South Asia, which is less than 1/6 of our figure for China's household coal burning. We also notice that the calculated BC emissions from household coal burning were 482 Gg in 2013, less than BrC emissions in the same period, suggestive of the relatively high BrC emissions from China's residential coal burning and deserving special attention and efforts.

Questions may arise regarding the share of briquettes in the total household coal consumption. The percentage "20%" has been used for more than 10 years in our studies (e.g., Chen et al., 2005; Zhi et al., 2008). This percentage is now being seriously challenged by a more complicated situation of coal consumption in China's households. On the one hand, Chinese government has long since promoted the use of coal briquettes to achieve cleaner emission target, which helps increase the share of briquettes (Chen et al., 2015b); on the other hand, the increasing reliance on burning raw-chunks for room heating (through circulating hot-water) in northern China is ridding briquettes of, but bringing chunks into, households, which results in a declined briquette share (Zhi et al., 2017). As a result, it is difficult to establish whether the assigned "20%" is higher or

lower than the actual one, which adds uncertainty to the estimates of the emissions and optical effects for China's household coal burning.

The huge emissions of BrC from household coal burning suggest the importance of including BrC in calculations of the total light absorption of coal emissions for understanding the BrC-related global energy budget. Given the more established knowledge of BC's optical properties and climate consequences compared to BrC's, exploration of the light-absorbing relationship between BrC and BC helps to substantiate the importance of BrC in the current discussion of climate effects of light-absorbing carbonaceous aerosols. Here $f_{BrC}(\lambda)$ is used to quantitatively describe the fraction of BrC absorption in the combined light absorption of BrC+BC at each wavelength of the scanned solar spectrum (refer to Supporting Information for the method to calculate $f_{BrC}(\lambda)$). The results of $f_{BrC}(\lambda)$ were plotted in Figure 6 for coal briquette, coal chunk and the average over briquette and chunk weighted by their consumption shares. According to Figure 6, the values of $f_{BrC}(\lambda)$ for briquette, chunk, and the average all increase towards short wavelengths, directly corroborating the conventional understanding that the light absorption by BrC increases more than that by BC from the green to violet spectral ranges due to the stronger spectral dependence of absorption by BrC than that by BC (Hoffer et al., 2006; Chakrabarty et al., 2010; Kirchstetter et al., 2012).

Moreover, Figure 6 demonstrates a significantly higher $f_{BrC}(\lambda)$ for briquettes (green line) than for chunks (black line), which corresponds to a higher BrC/BC ratio for briquettes (7.70±3.28) than for chunks (1.45±0.68) (based on Table 2) and to a higher AAE for briquettes (2.55±0.44) than for chunks (1.30±0.32) (Figure 5). In consideration of the share of briquette or anthracite in the total residential coal consumption, the calculated average $f_{BrC}(\lambda)$ (red line) over all residential coal consumption (including bituminous coals and anthracites in either chunk or briquette styles) is found to range from 0.061 (at 850 nm) to 0.470 (at 355 nm). Integration of $f_{BrC}(\lambda)$ and solar spectrum results in $F_{BrC}$, the fraction of absorbed solar radiance by BrC relative to the total absorption (refer to Supporting Information for the method for calculating $F_{BrC}$). A value of 0.265 is obtained for $F_{BrC}$ for the wavelength range from 350 to 850 nm. This means that in the scenario of current household coal burning in China, solar light absorption by BrC accounts for more than a quarter of the total absorption, while the other 73.5% is attributable to BC. This implies that although BrC plays a less important role in solar light absorption than BC regarding light absorption by carbonaceous emissions from the residential sector, it is absolutely non-negligible. The recommendation of adding BrC to climate modeling merits serious consideration for better modeling-based climate predictions.

**Data availability**

The research data can be accessed on request to the corresponding author (zhigr@craes.org.cn).

**Acknowledgements**

5   This study was supported by the National Natural Science Foundation of China (41373131, 41173121).

*Competing interests.* The authors declare that they have no conflict of interest.

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

Table 1. Coals used in this study.

| Coal | $M_{ad}$[a] | $A_{ad}$[b] | $V_{daf}$[c] | $F_{Cad}$[d] | Rank[e] |
|------|------|------|------|------|------|
| NX | 1.00 | 17.59 | 7.61 | 74.22 | AN |
| CZ | 0.91 | 10.69 | 12.59 | 77.26 | SA |
| LL | 0.79 | 7.95 | 19.35 | 73.60 | LVB |
| PDS | 0.39 | 10.06 | 26.25 | 65.63 | MVB |
| SYS | 2.22 | 22.33 | 33.20 | 50.40 | HVB |
| XLZ | 2.70 | 12.77 | 38.58 | 51.92 | HVB |
| LK | 22.56 | 11.51 | 49.39 | 33.34 | HVB |

Notes:[a]Moisture on air-dry basis (%); [b]Ash on air-dry basis (%); [c]Volatile matter on dry and ash-free basis (%); [d]Fixed carbon on air-dry basis; [e] Rank by ASTM standard classification of coal [American Society for Testing and Material, 2004]. HVB is for high-volatile bituminous coal, MVB is for medium-volatile bituminous coal, LVB is for low-volatile bituminous coal, SA is for semi-anthracite, and AN is for ordinary anthracite. In addition, the 7 coals were produced in Ningxia Hui Nationality Autonomous Region (NX), Changzhi City of Shanxi Province (CZ), Lvliang City of Shanxi Province (LL), Pingdingshan City of Henan Province (PDS), Shuangyashan City of Heilongjiang Province (SYS), Xinglongzhuang Coal Mine of Shandong Province (XLZ), and Longkou City of Shandong Province (LK), respectively.

**Table 2. Measured emission factors (g/kg) of BrC and BC for China's household coal combustion.**

| Coal | Briquette in WJ stove | | Chunk in SC stove | | Chunk in HD stove | | Chunk in LW stove | | Average over Chunks | |
|---|---|---|---|---|---|---|---|---|---|---|
| | $EF_{BrC}$ | $EF_{BC}$ | $EF_{BrC}$ | $EF_{BC}$ | $EF_{BrC}$ | $EF_{BC}$ | $EF_{BrC}$ | $EF_{BC}$ | $EF_{BrC}$ | $EF_{BC}$ |
| **Anthracite** | | | | | | | | | | |
| NX | 1.31 | 0.095 | 0.12 | 0.036 | 0.39 | 0.16 | 0.93 | 0.55 | 0.51 | 0.26 |
| | 1.51 | 0.10 | 0.098 | 0.024 | 0.58 | 0.16 | 0.95 | 0.65 | | |
| CZ | 1.87 | 0.36 | 0.82 | 0.34 | 0.48 | 0.16 | 4.09 | 0.92 | 1.65 | 0.59 |
| | 1.40 | 0.28 | 1.43 | 0.47 | 0.45 | 0.16 | 2.61 | 1.50 | | |
| Mean of anthracites | 1.52 | 0.21 | 0.62 | 0.22 | 0.48 | 0.16 | 2.15 | 0.91 | 1.08 | 0.43 |
| Standard deviation | 0.16 | 0.16 | 0.72 | 0.27 | 0.01 | 0.00 | 1.70 | 0.43 | 0.80 | 0.23 |
| **Bituminous** | | | | | | | | | | |
| LL | 2.85 | 0.44 | 6.08 | 9.29 | 1.63 | 1.60 | 9.67 | 6.19 | 5.51 | 5.35 |
| | 2.10 | 0.38 | 4.24 | 7.22 | 2.28 | 1.32 | 9.15 | 6.48 | | |
| PDS | 8.13 | 0.75 | 9.41 | 13.93 | 5.28 | 2.17 | 9.98 | 10.72 | 8.69 | 10.15 |
| | 4.71 | 1.00 | 7.65 | 17.88 | 7.42 | 3.46 | 12.36 | 12.70 | | |
| SYS | 6.88 | 0.65 | 6.05 | 5.85 | 9.20 | 8.21 | 20.05 | 10.66 | 11.49 | 8.75 |
| | 5.81 | 0.71 | 6.04 | 4.07 | 10.02 | 6.30 | 17.55 | 17.34 | | |
| XLZ | 3.36 | 0.51 | 12.15 | 8.95 | 13.18 | 8.46 | 8.06 | 8.62 | 11.02 | 8.81 |
| | 2.53 | 0.53 | 14.14 | 10.30 | 10.69 | 7.09 | 7.92 | 9.47 | | |
| LK | 1.77 | 0.31 | 6.80 | 7.30 | 2.95 | 1.18 | 9.31 | 7.91 | 6.26 | 6.18 |
| | 1.97 | 0.31 | 5.24 | 7.38 | 3.07 | 1.54 | 10.01 | 11.80 | | |
| Mean of Bituminous coals | 4.01 | 0.56 | 7.78 | 9.21 | 6.59 | 4.14 | 11.41 | 10.20 | 8.59 | 7.85 |
| Standard deviation | 2.19 | 0.22 | 3.25 | 4.10 | 4.23 | 3.15 | 4.29 | 2.87 | 2.70 | 2.00 |

Notes: The 4 stoves are respectively Wanjia brand briquette stove (WJ), Simple Chunk stove (SC), Huanding brand chunk stove (HD), and Laowan brand chunk stove (LW).

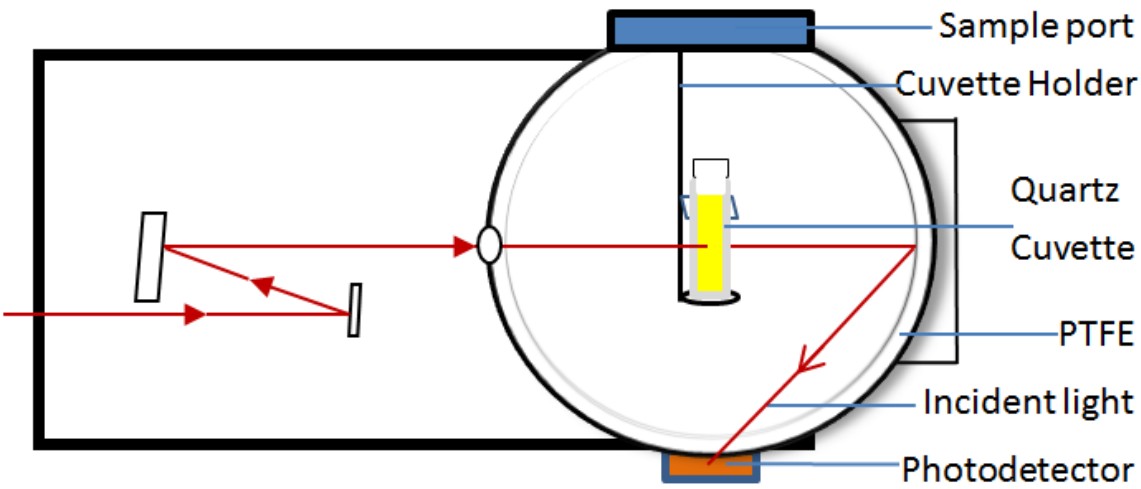

**Figure 1. The sketch of integrating sphere (IS) method.** PTFE means Polytetrafluoroethylene.

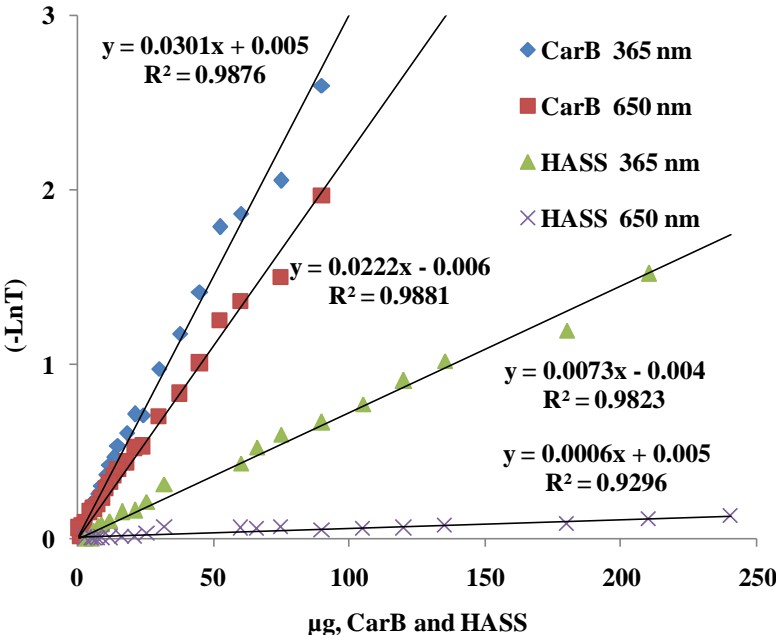

**Figure 2. The calibration curves for CarB (carbon black, diamonds, squares) and HASS (humic acid sodium salt, forks, triangles) at 365 and 650 nm.** T is the transmittance of incident light through calibration solution.

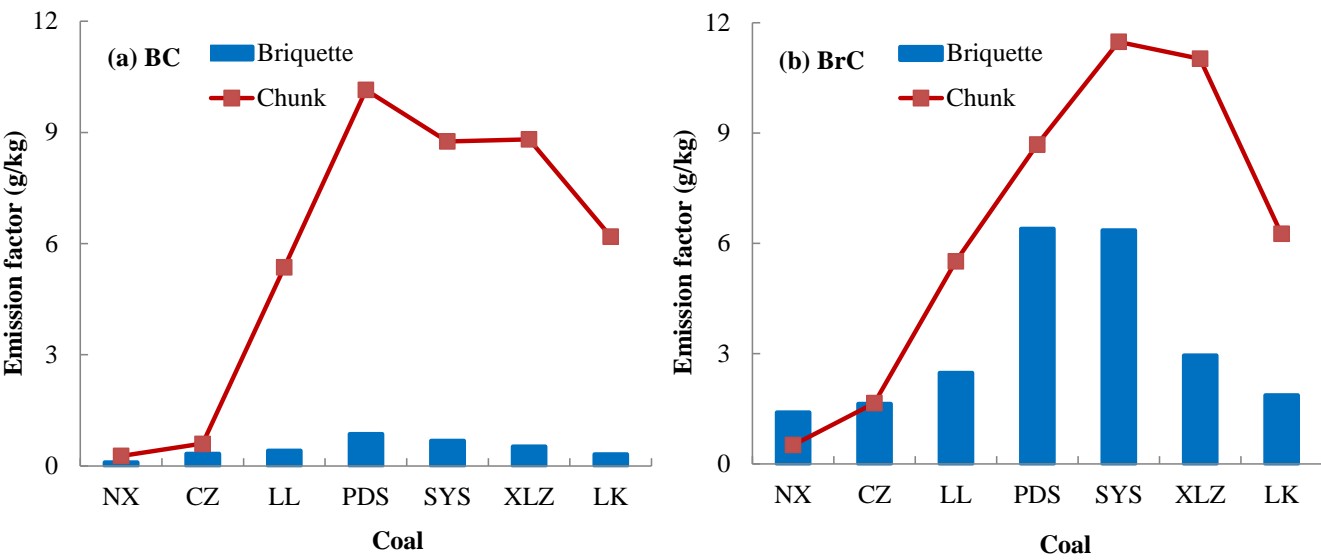

Figure 3. Effects of coal briquetting on black carbon (BC) and brown carbon (BrC) emissions.

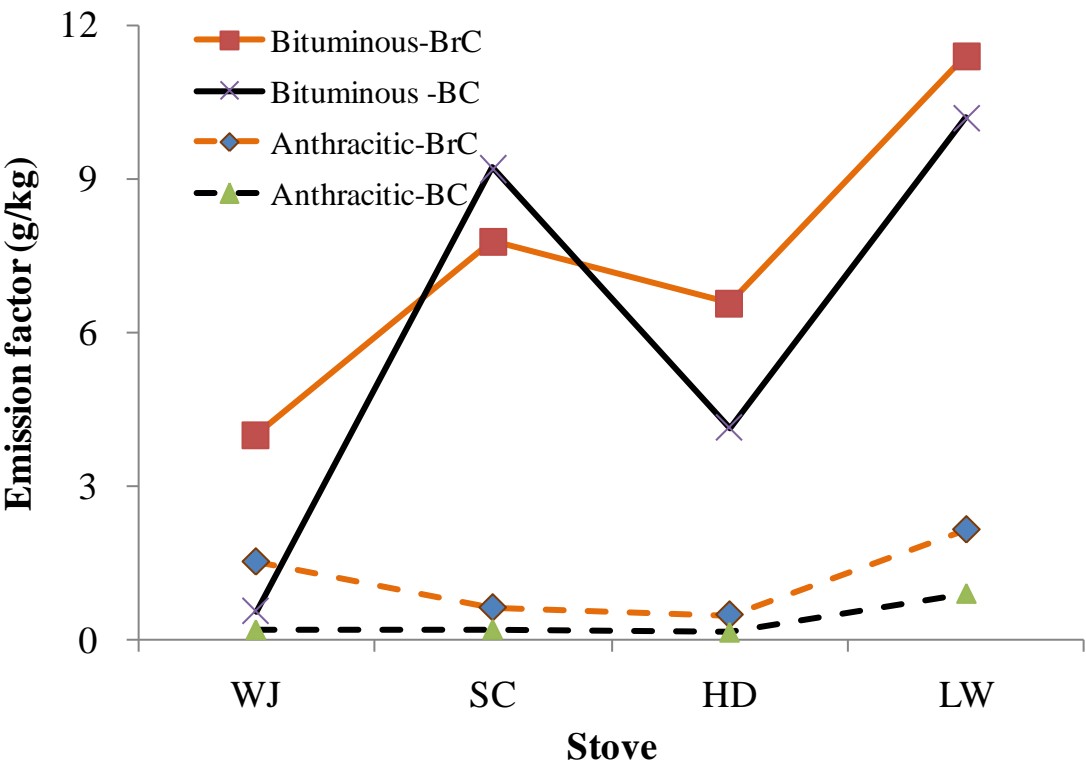

**Figure 4. Comparison of emission factors between bituminous and anthracitic coals.** The 4 stoves are respectively Wanjia brand briquette stove (WJ), Simple Chunk stove (SC), Huanding brand chunk stove (HD), and Laowan brand chunk stove (LW).

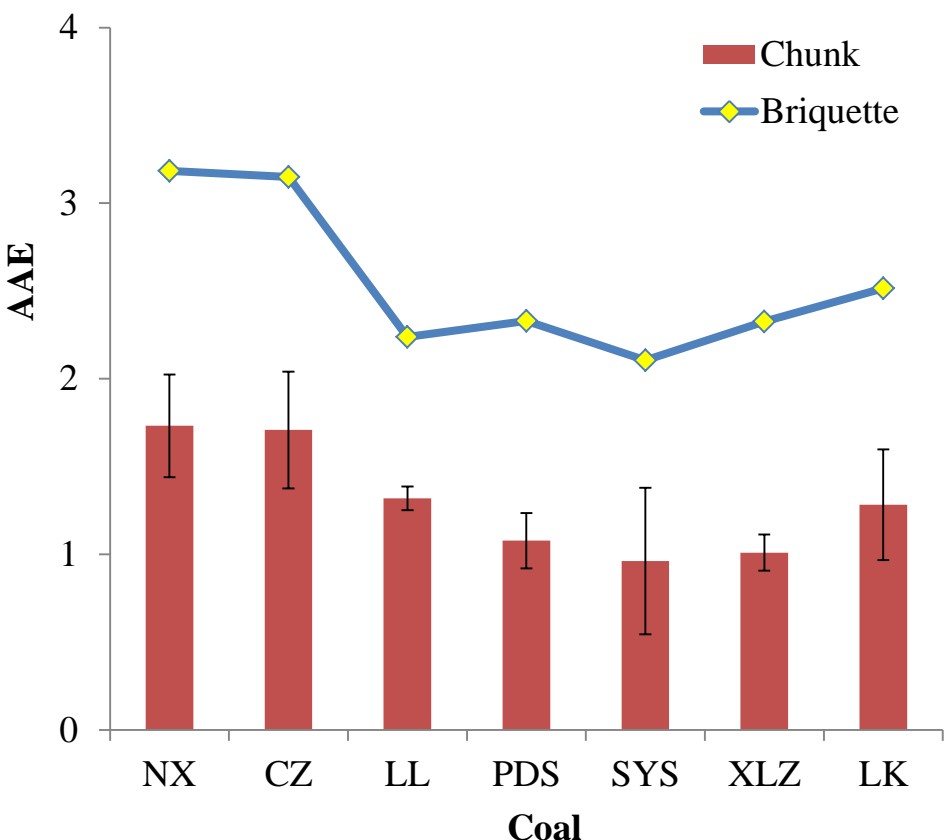

**Figure 5. Comparison of AAEs** (absorption Ångström exponents) **between briquette and chunk coals.**

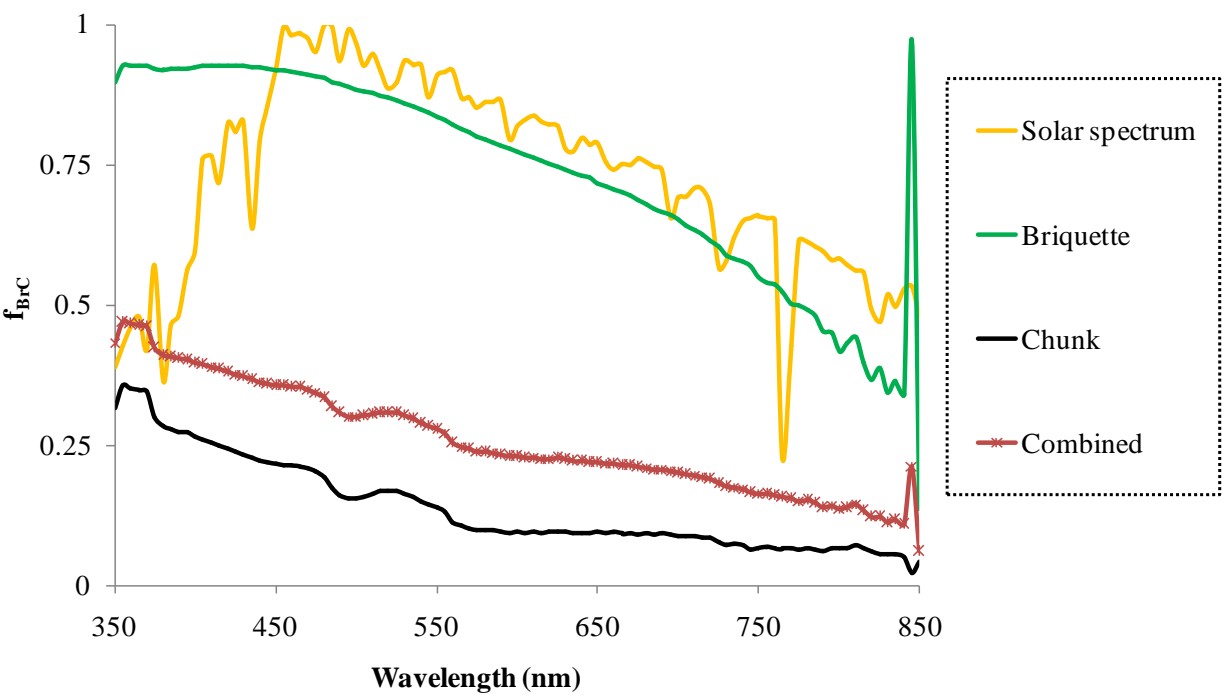

**Figure 6. Fraction of light absorption by BrC in total absorption by BrC+BC from residential coal burning.** The fraction is expressed as $f_{BrC}(\lambda)$ and was calculated in accordance with the method described in the Supporting Information. The yellow line is the clear sky air mass one global horizontal solar spectrum at the earth's surface in relative unit (Levinson et al., 2010).