# Peer review of "Emission factors and light absorption properties of brown carbon from household coal combustion in China"

_Atmospheric Chemistry and Physics, 2016_

## Referee Comment (RC1) · S. Mogo (Referee) · 19 Jan 2017

The manuscript "Emission factors and light absorption properties of brown carbon from household coal combustion in China" by Sun et al., presents valuable measurements of the emission factors and optical properties of brown carbon and black carbon fractions resulting from household coal burned for heating/cooking purposes in China. An optical method using an integrating sphere was applied to analyse several coals burned in typical stoves at both, chunk and briquette styles.

The method is not new but it is applied to an interesting study. The protocols are sufficiently well described, the study has been done carefully and the interpretation makes sense. The results are potentially interesting for other researchers engaged in
aerosol studies.

The paper is adequate to the journal scope, it is well formatted and presents a valuable experiment, so it deserves publication in the ACP. Except for some small technical corrections, I have only one major concern about reference Sun *et al.*, 2016. Please see bellow.

**Specific comments and technical corrections:**

- pag.5, line 17: L/min should be l/min;

- pag.5, line 18: "... into the PFS-4000..." should be "...into a PFS-4000...";

- pag.9, line 18: something is missing in the sentence starting with "It's interesting that...";

- please complete captions of figures and tables to make them more informative, for example, describing the meaning of the abbreviations used;

- reference Sun *et al.*, 2016 is an unpublished work and, based only on the title, it appears to have a significant overlap with the present manuscript, so authors should explain what part of the work is done in each manuscript.

---

## Referee Comment (RC2) · G. SHEN (Referee) · 1 Feb 2017

Black carbon (BC) and Brown carbon (BrC) have significant impacts on regional climate change and residential coal combustion is a major source of these pollutants. Emission data of BC and BrC from residential solid fuel combustion, depending on coals and stove types are still limited, though more and more studies are available during the last several years (most reported BC and/or elemental carbon). The present study measured emissions of BC and BrC from residential coal combustion in 7 different coals burnt in 4 different stoves in China. The study provided firsthand data on pollutant emission factors from residential coal combustion that would be interesting and helpful for emission inventory, air quality modelling and also control policies on raw

coal combustion in the country. Overall the experiments are well conducted and the manuscript is clearly present. I have the following concerns to be considered by the authors before the paper published in ACP.

1. The study used optical integrating sphere method to quantify BC and BrC amounts. The quantification is based on calibration curves using CarB and HASS as standards. Chemical compositions and properties could be different between BrC and HASS (Page 6 lines 20-25). In fact, these two terms (BC and BrC) are initially defined based on the light properties while chemical compositions of BrC is still not well characterized. So in my opinion although the difference here is out of this study scope it might be necessary and important to highlight in the paper that results reported here is probably different from others in literature reporting BC and/or BrC using other measurement techniques, which should be considered in the generation of results. What reported here are more likely a CarB-equivalent and HASS-equivalent carbon emissions. Interesting to know whether the authors had BC data from typical BC measurement instruments or BrC data from carbon analysis, and are the results comparable if you have the data?

2. Section 3.1-influence of briquetting, and Figure 3: are the results for chunk coals from traditional SC stove, or averaged from all three chunk stoves? Note that briquettes were burned in WJ stove while raw chunk coals were burned in another three stoves, therefore differences observed here are resulting from not only coal briquetting but also stove types. Can the author evaluate difference among different stoves, for example, emissions from chunk between traditional SJ stove and HD/LW stoves?

3. As only two repeats for each experiment, it is inappropriate to calculate standard deviations. In Table 2, suggest to put all data for each coal. The overall means with standard deviations for anthracite and bituminous are fine. Also replace means with standard deviations by ranges in text if only two numbers.

4. As the research group had published a series of BC emission data from coal combustion, it is suggested to compare and discuss the results here and the previous studies, taking differences (if any) of coal types and stoves in the consideration. Will the overall means of BC for anthracite and briquettes change when taking more data into account, and how much the variation would be?

5. Lab experiments are easier to be controlled and repeatable, but more and more studies suggested that lab studies may fail to simulate high emissions and be difficult to capture high variations in real field. Both methods have advantage and disadvantage. The limitation and consequent impacts on the generation of these lab-experimental data should be briefly discussed in main text.

some specific ones:

Page 5 line 23, "PFS-4000" not "FPS"

Page 7 line 10-15, the information should be shortened and moved to the section 2.4.

Page 8 line 25-30, high emissions from medium-volatile content coals are also found in PAHs emissions. Therefore, it appears that this type of coals should be eliminated in use.

Page 9 line 11, delete "either"

Page 9 line 25, please note that 20% briquettes probably underestimated, though no reliable statistical data available so far. Suggest to add a discussion on uncertainties of these estimated emissions, due to fractions of briquettes and variations of EFs

Page 9 line 30, "less than BrC emissions from residential coal combustion in the same period"?

Table 1, can the authors provide heating values of these coals? And, are these properties like moisture and ash content changed when briquetting?

[Figure]

---

## Author Comment (AC1) · 14 Mar 2017

**Reply to Referee 1[#]**

**General comment:**

The manuscript "Emission factors and light absorption properties of brown carbon from household coal combustion in China" by Sun et al., presents valuable measurements of the emission factors and optical properties of brown carbon and black carbon fractions resulting from household coal burned for heating/cooking purposes in China. An optical method using an integrating sphere was applied to analyze several coals burned in typical stoves at both chunk and briquette styles. The method is not new but it is applied to an interesting study. The protocols are sufficiently well described, the study has been done carefully and the interpretation makes sense. The results are potentially interesting for other researchers engaged in aerosol studies. The paper is adequate to the journal scope, it is well formatted and presents a valuable experiment, so it deserves publication in the ACP. Except for some small technical corrections, I have only one major concern about reference Sun et al., 2016. Please see below.

**Response:**

Thanks a lot for the positive comments and recommendation for publication in ACP. We noticed that the reviewer has only one major concern, which is on the reference "Sun et al., 2016". We will specifically address such concern in our response to "**Comment 5**" below.

**Comment 1:**

Page 5 line 17: L/min should be l/min;

**Response:**

We have changed "L/min" to "l/min" in revised version (page 5 line 17).

**Comment 2:**

Page 5 line 18: "... into the PFS-4000..." should be "...into a FPS-4000...";

**Response:**

Thanks for reminder. We have changed "into the PFS-4000" to "into a FPS-4000" in revised version (page 5 line 18).

**Comment 3:**

Page 9 line 18: something is missing in the sentence starting with "It's interesting that...";

**Response:**

The sentence is a complex sentence with "it" as the formal subject and the "that-clause" as the logic subject (page 10 line 12, revised version). With this sentence, we intend to show that the absorption Ångström exponent (AAE) comes lowest in the coals of medium $V_{daf}$, which happens to be opposite to that $EF_{BC}$ and $EF_{BrC}$ comes highest in the coals of medium $V_{daf}$. This can be seen by

comparison of our two figures in the manuscript (Figure 3, Figure 5). In fact, some of our previous studies (e.g., Chen et al., 2006; Zhi et al., 2008, 2009) have repeatedly concluded that the maximal $EF_{BC}$ occurs in medium maturity coals around $V_{daf} = 30\%$.

[Figure]

Fig. 3. "bell-shape" for $EF_{BC}$ against $V_{daf}$  Fig. 5. "U-shape" for AAE against $V_{daf}$

**Comment 4:**

Please complete captions of figures and tables to make them more informative, for example, describing the meaning of the abbreviations used;

**Response:**

Thank you. We have examined all figures and tables and tried to make their captions more informative by explaining the meaning of the abbreviations, as follows:

(1) Page 21 revised version, Table 1: We added the full names of the coal mine locations of the 7 coals (at the end of the table).

(2) Page 22 revised version, Table 2: We added the full names of the 4 coal stoves (at the end of the table).

(3) Page 23 revised version, Figure 1: add "PTFE means Polytetrafluoroethylene" to the caption.

(4) Page 24 revised version, Figure 2: change "CarB (diamonds, squares) and HASS (forks, triangles)" to "CarB (carbon black, diamonds, squares) and HASS (humic acid sodium salt, forks, triangles)".

(5) Page 25 revised version, Figure 3: "BC and BrC" have changed to "black carbon (BC) and brown carbon (BrC)".

(6) Page 26 revised version, Figure 4: add "The 4 stoves are respectively Wanjia brand briquette stove (WJ), Simple Chunk stove (SC), Huanding brand chunk stove (HD), and Laowan brand chunk stove (LW)" to the caption.

(7) Page 27 revised version, Figure 5: change "of AAEs" to "of AAEs (absorption Ångström exponents)" in the caption.

**Comment 5:**

Reference Sun et al., 2016 is an unpublished work and, based only on the title; it appears to have a significant overlap with the present manuscript, so authors should explain what part of the work is done in each manuscript.

**Response:**

The reference "Sun et al., 2016" is indeed our unpublished work. Different from the present work

that focuses on household coal in terms of BrC emissions, "Sun et al., 2016" had intended to focus on household biomass in terms of BrC emissions. The same method (i.e., the integrating sphere, IS) had been planned for these twin papers. However the "Sun et al., 2016" paper (on biomass) has so far not been finished and submitted, which makes our reference senseless. In this case we have to cancel the reference to "Sun et al., 2016" throughout our revised version.

By the way, the planned paper "Sun et al., 2016" is close to being finished and will in return refer to the current paper (on coal) regarding IS method.

Thanks again for the careful and constructive reviewing.

**References**

Chen, Y., Zhi, G., Feng, Y., Fu, J., Feng, J., Sheng, G., and Simoneit, B. R. T.: Measurements of emission factors for primary carbonaceous particles from residential raw-coal combustion in China, Geophys. Res. Lett., 33 (20), 1-4, 10.1029/2006gl026966, 2006.

Zhi, G., Chen, Y., Feng, Y., Xiong, S., Li, J., Zhang, G., Sheng, G., and Fu, J.: Emission characteristics of carbonaceous particles from various residential coal-stoves in china, Environ. Sci. Technol., 42 (9), 3310-3315, 10.1021/es702247q, 2008.

Zhi, G., Peng, C., Chen, Y., Liu, D., Sheng, G., and Fu, J.: Deployment of coal briquettes and improved stoves: possibly an option for both environment and climate, Environ. Sci. Technol., 43 (15), 5586-5591, 10.1021/es802955d, 2009.

---

## Author Comment (AC2) · 14 Mar 2017

**Reply to Referee 2[#]**

**General Comment:**

Black carbon (BC) and Brown carbon (BrC) have significant impacts on regional climate change and residential coal combustion is a major source of these pollutants. Emission data of BC and BrC from residential solid fuel combustion, depending on coals and stove types are still limited, though more and more studies are available during the last several years (most reported BC and/or elemental carbon). The present study measured emissions of BC and BrC from residential coal combustion in 7 different coals burnt in 4 different stoves in China. The study provided firsthand data on pollutant emission factors from residential coal combustion that would be interesting and helpful for emission inventory, air quality modelling and also control policies on raw coal combustion in the country. Overall the experiments are well conducted and the manuscript is clearly present.

**Response:**

Many thanks for the positive comments on the significance of our research and the quality of our manuscript. We would further improve our manuscript according to the comments of reviewer.

**Comment 1:**

The study used optical integrating sphere method to quantify BC and BrC amounts. The quantification is based on calibration curves using CarB and HASS as standards. Chemical compositions and properties could be different between BrC and HASS (Page 6 lines 20-25). In fact, these two terms (BC and BrC) are initially defined based on the light properties while chemical composition of BrC is still not well characterized. So in my opinion although the difference here is out of this study scope it might be necessary and important to highlight in the paper that results reported here is probably different from others in literature reporting BC and/or BrC using other measurement techniques, which should be considered in the generation of results. What reported here are more likely a CarB-equivalent and HASS-equivalent carbon emissions. Interesting to know whether the authors had BC data from typical BC measurement instruments or BrC data from carbon analysis, and are the results comparable if you have the data?

**Response**:

(1) Integrating Sphere (IS) method was utilized in this study to separate the contributions of BrC and BC in terms of light absorption. Considering that preferred reference materials were often carbon black (CarB) for BC and humic acid sodium salt (HASS) for BrC (Heintzenberg, 1982; Hitzenberger et al., 2006; Reisinger et al., 2008; Wonaschütz et al., 2009), we carried over this philosophy to the current study. This implies our consent to the assumption that BC and BrC in our samples collected for household coal smoke have the same light-absorbing properties of CarB and HASS, respectively and what reported here are essentially CarB-C-equivalent and HASS-C-equivalent. Consequently it is not surprising that the results obtained here are probably different from others in literature reporting BC and/or BrC using

other measurement techniques (e.g., thermal/optical method or aethalometer) (Chen et al., 2006; Zhi et al., 2008, 2009; Shen et al., 2013, 2014; Aurell and Gullett, 2013) or reference materials (e.g., Fulvic acid, humic acid or HULIS) (Duarte et al. 2007; Lukács, et al. 2007; Baduel et al. 2009, 2010), in view of both BC and BrC being method-defined.

In the revised version, we highlighted above opinion (page 6 lines 27-30).

(2) We noticed the reviewer's interest in whether we had BC data from typical BC measurement instruments or BrC data from carbon analysis, and whether the results are comparable. Actually in the current study we have both BC data from thermal/optical reflectance (TOR) carbon analysis method (expressed as "elemental carbon", EC) and BC data from IS method. BC-EC paired data are given here as well as in the Supporting Information (see Table S1 and Figure S3, data are for coal briquette in WJ stove and for coal chunk in averaged 3 stoves, SC, HD, and LW). Although IS-BC is somewhat higher than TOR-EC in most cases, they are correlated significantly.

We briefly describe this relationship in our revised manuscript (page 8 lines 20-22 in subsection 3.1).

**Table S1. Comparison between IS-based $EF_{BC}$ and TOR-based $EF_{EC}$**

| Coal | Chunk (Average over 3 Chunk stoves) | |
|------|-----------|-----------|
| | $EF_{BC}$ | $EF_{EC}$ |
| NX | 0.26 | 0.12 |
| CZ | 0.59 | 0.29 |
| LL | 5.35 | 4.25 |
| PDS | 10.15 | 8.09 |
| SYS | 8.75 | 10.50 |
| XLZ | 8.81 | 6.80 |
| LK | 6.18 | 4.06 |
| | **Briquette (WJ stove)** | |
| NX | 0.10 | 0.08 |
| CZ | 0.32 | 0.07 |
| LL | 0.41 | 0.27 |
| PDS | 0.86 | 0.41 |
| SYS | 0.68 | 0.80 |
| XLZ | 0.52 | 0.71 |
| LK | 0.31 | 0.21 |

[Figure]

**Figure S3.    The correlation of EFs between IS-based $EF_{BC}$ and TOR-based $EF_{EC}$**

**Comment 2:**

Section 3.1 influence of briquetting, and Figure 3: are the results for chunk coals from traditional SC stove, or averaged from all three chunk stoves? Note that briquettes were burned in WJ stove while raw chunk coals were burned in another three stoves, therefore differences observed here are resulting from not only coal briquetting but also stove types. Can the author evaluate difference among different stoves, for example, emissions from chunk between traditional SC stove and HD/LW stoves?

**Response:**

The reviewer's description is right. In Section 3.1 and Figure 3, the emission factors (BC and BrC) for briquettes are the results from WJ stove (for briquette only), while those for chunks are the results from averaging over another three chunk-stoves (SC, HD, and LW). Because the WJ stove burned briquettes only and all the 3 chunk stoves burned coal chunks only, it is improper to compare the EFs between WJ and the other 3 chunk stoves. However, it is reasonable if we compare EFs among the three chunk stoves (SC/HD/LW). We take the advantage of Figure 4, finding that, among the 3 chunk stoves, both $EF_{BrC}$ and $EF_{BC}$ of coal chunks are in the order of LW>SC>HD, regardless of bituminous coal or anthracite coal, reflecting that HD stove performs the best and the LW stove, the worst. As mentioned in subsection 2.1, stove SC is of traditional style widely used especially in past decades in China's households for heating rooms through direct radiation of coal-burning whereas HD and LW are actually household mini-boilers of low pressure type used for heating rooms by a water piping/radiating system. The above order of LW>SC>HD implies that current transfer from stove type of direct coal combustion radiation (e.g., SC stove) to that of heated water piping/radiating system (e.g., LW and HD stoves) in households does not necessarily leads to a decline of EFs.

In the revised version, we described above notion briefly in subsection 3.2 (page 9 lines 14-21).

**Comment 3:**

As only two repeats for each experiment, it is inappropriate to calculate standard deviations. In Table 2, suggest to put all data for each coal. The overall means with standard deviations for anthracite and bituminous are fine. Also replace means with standard deviations by ranges in text if only two numbers.

**Response:**

We accept the suggestion and put all data we have got for each coal into Table 2. We also replaced "mean±standard deviation" manner in text with "ranges" manner (revised version, subsection 3.2 page 9 lines 8-10).

**Comment 4:**

As the research group had published a series of BC emission data from coal combustion, it is suggested to compare and discuss the results here and the previous studies, taking differences (if any) of coal types and stoves in the consideration. Will the overall means of BC for anthracite and briquettes change when taking more data into account, and how much the variation would be?

**Response:**

Thanks for the knowledge of a series of our previous studies of BC emission data from coal combustion. It is a good idea to compare and discuss the results of this study and the previous ones, taking differences of coal types and stoves in the consideration. We brought all of the directly measured BC emission factors in our previously published articles (Chen et al., 2005, 2006, 2015; Zhi et al., 2008,

2009) into the following Table S2, together with the emission factors measured in this study. Comparison between previous and current studies shows that the means of the emission factors for either anthracite coals or bituminous coals in either briquette or chunk styles in this study are somewhat higher than those in previous ones. However the key findings in previous studies still stand in this study; for example, bituminous coals or raw chunks usually release more pollutants (including BC) than anthracites or briquettes, and BC emission factor usually peaks in medium-volatile bituminous coal, etc. The differences in reported EFs are generally from a variety of factors, such as stoves, briquetting procedures, combustion manipulations, and even the quantification methods (Chow et al., 2001; Zhi et al., 2009, 2011). As an example, in our response to Comment 1, we showed that IS-BC is generally higher than TOR-EC. Moreover, we know TOR-EC is almost always higher than thermal/optical transmittance EC (TOT-EC) (Chow et al., 2001; Zhi et al., 2011) and our EC values in our previous studies were mostly originated from TOT protocol (NIOSH), which further enlarges the difference between this study and previous ones.

**Table S2**. **The collection of the directly measured BC (EC) emission factors in our previous articles**

| | | Anthracite | | Bituminous | | | |
| --- | --- | --- | --- | --- | --- | --- | --- |
| | | **Raw chunk** | **Briquette** | **Raw chunk** | | **Briquette** | |
| 1 | Chen et al, 2005 | | 0.004 | | | 0.096 | 0.675 |
| | | | | | | 0.523 | 0.064 |
| 2 | Zhi et al, 2009 | 0.035 | 0.012 | 0.13 | 0.73 | 0.009 | 0.014 |
| | | 0.005 | 0.001 | 16.9 | 10.3 | 0.076 | 0.016 |
| | | | | 28.5 | 4.35 | 0.080 | 0.034 |
| | | | | 1.48 | | 0.019 | |
| 3 | Chen et al,2007 | 0.007 | | 0.20 | 5.34 | | |
| | | 0.002 | | 10.10 | 10.12 | | |
| | | | | 12.67 | 6.97 | | |
| | | | | 0.48 | | | |
| 4 | Chen et al, 2016 | 0.02 | 0.06 | 1.71 | | 0.67 | |
| | | | | 2.38 | | 1.3 | |
| | | | | 3.46 | | 1.28 | |
| | | | | 0.61 | | 0.07 | |
| 5 | Zhi et al, 2010 | | | 0.042 | 0.51 | 0.011 | 0.054 |
| | | | | 1.23 | 2.89 | 0.085 | 0.16 |
| | | | | 0.18 | 0.083 | 0.034 | 0.018 |
| | | | | 0.23 | 4.83 | 0.044 | 0.52 |
| | | | | 10.02 | 11.17 | 0.64 | 0.47 |
| | | | | 5.77 | 0.55 | 0.31 | 0.084 |
| **Mean** | | **0.014** | **0.019** | **5.34** | | **0.27** | |
| **sd.** | | **0.014** | **0.028** | **6.36** | | **0.37** | |
| 6 | This study | 0.26 | 0.1 | 5.35 | 10.15 | 0.41 | 0.86 |
| | | 0.59 | 0.32 | 8.75 | 8.81 | 0.68 | 0.52 |
| | | | | 6.18 | | 0.31 | |
| **Mean** | | **0.43** | **0.21** | **7.85** | | **0.56** | |
| **sd.** | | **0.23** | **0.16** | **2** | | **0.22** | |

Because our manuscript focuses on BrC instead of BC, we put this table in our Supporting

Information (Table S2) and meanwhile add a new paragraph in our revised version (subsection 3.2, page 9 lines 22-29).

**Comment 5:**

Lab experiments are easier to be controlled and repeatable, but more and more studies suggested that lab studies may fail to simulate high emissions and be difficult to capture high variations in real field. Both methods have advantage and disadvantage. The limitation and consequent impacts on the generation of these lab-experimental data should be briefly discussed in main text.

**Response:**

We totally agree to the comment on lab experiments and field tests. Generally laboratory experiments allow investigators to repeat the process under controlled conditions for investigating the effects of a specific influencing factor on the emissions (Jenkins et al., 1996; Roden et al., 2009; Zhang et al., 2011; Jetter et al., 2012). In this study, with lab experiment methodology, we could test the briquetting effects or coal rank effects by fixing some other conditions (using the same 7 coals, identical combustion manipulation, and consistent sampling system). However this is anyway not so realistic as in field circumference with random conditions that are difficult to reproduce in laboratory (Roden et al., 2006; Johansson et al., 2008; Christian et al., 2010). Just as the reviewer mentioned, "Both methods have advantage and disadvantage", depending on what purpose you prefer to pursue.

In our revised manuscript we described the limitation and consequent impacts on the generation of these lab-based data in the subsection 2.2 (page 5 lines 32-33 and page 6 lines 1-4).

**Some specific comments:**

**Comment 6:**

Page 5 line 23, "PFS-4000" not "FPS"

**Response:**

FPS is the acronym of "Fine Particle Sampler". There are only 2 cases in our manuscript and both of them exist in subsection 2.2 (page 5 lines 18, 22, revised version).

**Comment 7:**

Page 7 line 10-15, the information should be shortened and moved to the section 2.4.

**Response:**

The paragraph in page 7 lines 11-14 (initial version) has been deleted. This means the second paragraph in the initial version would become the first paragraph in the new version. We added a new sentence to begin this paragraph, as "The calculated emission factors of BrC and BC for the coal/stove combinations are presented in Table 2" (page 7 line 19, revised version).

**Comment 8:**

Page 8 line 25-30, high emissions from medium-volatile content coals are also found in PAHs emissions. Therefore, it appears that this type of coals should be eliminated in use.

**Response:**

It is true that high emissions from medium-volatile content coals were also found in PAHs emissions (Shen et al., 2013). We added a sentence in subsection 3.2 at page 9 lines 12-13 (revised version).

**Comment 9:**

Page 9 line 11, delete "either"

**Response:**

Thanks for reminder. The "either" in our initial submission is useless and is due to carelessness. We deleted it in our revised version (Page 10 line 5).

**Comment 10:**

Page 9 line 25, please note that 20% briquettes probably underestimated, though no reliable statistical data available so far. Suggest to add a discussion on uncertainties of these estimated emissions, due to fractions of briquettes and variations of EFs.

**Response:**

In terms of the share of briquettes in total household coal consumption, the percentage "20%" has been used for more than 10 years (Chen et al., 2005; Zhi et al., 2008). This percentage is now seriously challenged by a more complicated situation in China. On the one hand, Chinese government has long since promoted the use of coal briquettes to achieve cleaner emission target, which helps increase the share of briquettes (Chen et al., 2015); on the other hand, the increasing reliance on burning raw-chunks for room heating (through circulating hot-water) in northern China is ridding briquettes of but bringing chunks into households, which results in a declined briquette share (Zhi et al., 2017). As a result, it is difficult to establish whether the assigned "20%" in this study is higher or lower than the actual one, which adds uncertainty to the estimates of the emissions and optical effects for China's household coal burning.

Thus, we kept unchanged the "20%" for China's briquette share in household coal total and added a new paragraph for the discussion of uncertainty (subsection 3.4, page 10 lines 26-33, page 11 lines 1-2).

**Comment 11:**

Page 9 line 30, "less than BrC emissions from residential coal combustion in the same period"?

**Response:**

Yes, it is true that "the calculated BC emissions from household coal burning were…less than BrC emissions in the same period" (revised version, page 10 lines 22-24). This results from the higher $EF_{BrC}$ than $EF_{BC}$ (See Table 2 in subsection 3.1). However, it doesn't follow that BrC is more potent in light-absorption than BC in view of the far lower light absorption efficiency for BrC than for BC. In the subsection 3.4, we revealed that "in the scenario of current household coal burning in China, solar light absorption by BrC accounts for more than a quarter of the total absorption, while the other 73.5% is attributable to BC".

**Comment 12:**

Table 1, can the authors provide heating values of these coals? And, are these properties like moisture and ash content changed when briquetting?

**Response:**

Regrettably we don't have the heating values of coals used in the experiment. The properties like moisture and ash content were unavoidably changed by briquetting. We actually tried to equal the changes in moisture and ash content to even up the impacts of briquetting on EFs.

Thanks again for the careful and constructive reviewing.

[revised manuscript text omitted]